# The Role of Neuroticism in Predicting Psychological Harassment in Nursing: A Longitudinal Study

**DOI:** 10.3390/ijerph16050889

**Published:** 2019-03-11

**Authors:** Joana Fornés-Vives, Dolores Frias-Navarro, Gloria García-Banda, Marcos Pascual-Soler

**Affiliations:** 1Department of Nursing and Physiotherapy, University of Balearic Islands. Ctra Valldemossa km 7.5, Palma de Mallorca 07122, Spain; 2Department of Methodology of the Behavioural Sciences, University of Valencia. Av. de Blasco Ibáñez, 21, Valencia 46010, Spain; M.Dolores.Frias@uv.es; 3Department of Psychology, University of Balearic Islands. Ctra Valldemossa km 7.5, Palma de Mallorca 07122, Spain; ggbanda@uib.es; 4ESIC Business & Marketing School. Av. de Blasco Ibáñez, 55, Valencia 46021, Spain; marcos.pascual@esic.edu

**Keywords:** behavioral coping, emotional coping, harassment, neuroticism, nursing, personality

## Abstract

Psychological harassment is a serious occupational risk for nurses, but little is known about its related factors and possible predictors. The objective of the present study was to investigate whether nursing students’ neuroticism trait and coping styles can predict psychological harassment at work when they later become nurses. A non-experimental, longitudinal, three-wave prospective study with a time lag of 6 years was carried out, following nursing students from three Spanish universities until they joined the health labor market. The age range of the sample was 20–48 years, and the mean age was 26.99 ± 5.72; the majority of the sample were women (88.5%). Predictor variables were neuroticism and coping styles (emotional and behavioral coping). The criterion variable was psychological harassment. To examine the model fit between the predictor and criterion variables, we conducted structural equation modelling. Results confirmed a high correlation between neuroticism and psychological harassment. In addition, a direct effect of neuroticism on psychological harassment was found; however, emotional and behavioral coping styles did not show a good fit. Proactive interventions to improve emotional self-control are needed in order to prevent negative effects of psychological harassment at work on nurses.

## 1. Introduction

The European Foundation for the Improvement of Living and Working Conditions has shown that workplace violence (bullying, mobbing, psychological harassment), which can include verbal and physical abuse, is strongly associated with work-related stress and poor mental health [1]. Leymann defined this type of violence as mobbing, consisting of long-lasting, systematic psychological harassment from superiors, co-workers, or subordinates in order to victimize, humiliate, or threaten someone in the work setting [2]. Although this phenomenon is one of the main psychosocial risk factors at work, it could be based on previous learning styles [3]. In any case, its effective treatment is the responsibility of workers and organizations.

Research has identified factors associated with work-stress, bullying, and psychological harassment in the workplace [4]. However, less is known about the possible predictors, such as the stress involved in the transition from the student role to the nursing role [5] or the relationship between personality and bullying [6]. In addition, studies show that nurses they usually suffer from stress [7] and bullying [8] at work. A recent study pointed out that 30.2% of experienced nurses reported experiencing psychological harassment in their work places, from occasionally (17.2%), a few times per week (9.9%), to almost daily (1.1%) in the previous six months [9]. This violence has negative effects on nurses’ mental health [10]. 

Psychological harassment can be considered a work-related process which is influenced by both environmental and personal factors that can act independently of each other [6]. Among the personal factors, emotional instability or high neuroticism [6,11,12] and coping strategies [13] have been highlighted. Neuroticism is a personality trait that predisposes a person to experiencing feelings of emotional instability, such as anxiety, irritability, or impulsivity. In addition, people with some personality patterns, such as emotional instability, have been found to be more highly prone to maladjusted coping behaviors (e.g., addictions) [14]. People with higher scores on neuroticism are more likely to interpret ordinary situations as threatening and experience more stress [15]. The diathesis-stress model argues that some individuals’ biological predispositions make them more vulnerable to the effects of stressful life events [16]. From this perspective, the personal characteristics of individuals high in neuroticism would make them more prone to stress. Some studies have suggested that individuals with high neuroticism employ less adaptive strategies to deal with stress (more emotional coping than behavioral coping), such as escape, withdrawal, and self-blame [17], and there is evidence that the neurotic trait correlates positively with emotional coping [18,19]. Moreover, the neurotic trait and emotional coping have been shown to predict chronic stress, such as burnout, in nurses [20]. Identifying individual characteristics associated with harassment at work would provide important information for developing appropriate interventions to reduce chronic stress and enhance nurses’ wellbeing. 

Although the effect of personality and environmental factors on workplace violence has been explored, to our knowledge, no longitudinal studies have been published about the influence of the personality on psychological harassment, following a cohort of nurses from the beginning of their studies until their professional role. The present study addresses this gap. Based on the diathesis-stress model, we hypothesized that an emotional dispositional personality trait (high neuroticism) and certain styles of coping with stressful situations could increase vulnerability to harmful stress situations (harassment) in the workplace. More specifically, we hypothesized that:1The neuroticism personality trait, measured at T1, T2, and T3, positively predicts harassment at T3.2Emotional- and behavioral-focused coping, measured at T1, T2, and T3, predict harassment at T3.

## 2. Materials and Methods

A longitudinal, three-wave prospective study (T1-2007, start of academic studies *n* T1 = 249, T2-2010, end of academic studies *n* T2 = 199, T3-2013, after three years of work *n* T3 = 70) was carried out, following Spanish nursing students until they became registered nurses. At T3, the sample consisted of 70 nurses (10 men and 60 women) from three Spanish universities, with a mean age of 26.99 ± 5.72 years. For a more detailed description of the initial sample, see references [18,19].

### 2.1. Instruments

We measured personality scores with the Spanish version of the 12-item neuroticism-subscale of the NEO Five-Factor Inventory (NEO-FFI) [15]. The items measure the tendency to experience negative affect (e.g., “I seldom feel nervous”. Coping style scores were measured with two scales from the COPE Questionnaire adapted to the Spanish population [21]: “behavior-focused coping” (11 items: e.g., “I draw up an action plan”) and “emotion-focused coping” (12 items: e.g., “I get angry and let my emotions surface”). Psychological harassment was measured with the 35 items on the Psychological Harassment from the Workplace–Revised Version (HPT-R) questionnaire (e.g., “Expose you to group criticism”) [22]. Furthermore, we used three dichotomous (yes/no) final questions from the HPT-R questionnaire, designed to: (1) assess bullying using Leymann’s criterion (“Have you been subjected to any of the above acts at least once a week for at least 6 months?”); (2) identify possible witnesses to the harassment (“Have you been a witness to any of the acts mentioned above in your workplace?”); and (3) find out whether the participants currently feel psychologically harassed (“Do you currently think you are being psychologically harassed at work?”). Cronbach’s alphas for the instruments in our study were NEO-FFI = 0.86, behavior-focused coping = 0.67, emotion-focused coping = 0.85, and HPT-R = 0.86.

### 2.2. Procedure

The study was conducted in accordance with the Declaration of Helsinki. The procedure was approved by the Research Committee of the University of the Balearic Islands. Participation was voluntary. It was made clear that the participants could withdraw at any time and that the information would remain confidential. A full explanation of the research was provided, and informed consent was obtained. At T1 (start of nursing studies) and T2 (the end of the studies), the questionnaires were filled out in a regular lecture class. At T3 (three years after graduation), T2 participants were contacted by e-mail and encouraged to participate in the last wave of the study.

### 2.3. Data Analysis

Descriptive analyses were conducted using IBM SPSS v.24 (SPSS Inc., Chicago, IL, USA). After finding no attrition bias on the variables and between participants who remained and those who dropped out, we applied structural equation modelling (SEM) to test the hypotheses. SEM was calculated with EQS 6.2 software (Multivariate Software Inc., Encino, CA, USA) [23], using the covariance matrix and maximum likelihood method [23,24,25]. Model fit to the data was evaluated using the chi-square-χ^2^ statistic [26] which should not be statistically significant, and other fit indexes: Comparative Fit Index-CFI [27], Normed Fit Index-NFI [28], Non-Normed Fit Index-NNFI [29], whose value must exceed 0.90; and Standardized Root Mean Square Residual (SRMR), with a value below 0.06. 

## 3. Results

Descriptive analysis showed that neuroticism scores remained stable over time, and the use of emotional coping was predominant in all the assessments. In the case of reported psychological harassment measured at T3, participants retrospectively reported its occurrence in the previous two years. Thus, 8.8% of the participants perceived themselves as being harassed at least once a week for more than 6 months, 27.9% had witnessed harassment, and 5.9% felt harassed at the time of the assessment. The harassment could come from co-workers, supervisors, or subordinates, indistinctly.

Correlation analysis showed that psychological harassment was positively and significantly associated with neuroticism (*p* < 0.01) and coping styles (*p* < 0.05) in the different waves (Table 1). 

Following the criteria proposed by Hooper et al. [29] to test the main hypotheses, the SEM approach only showed a good approximate fit for hypothesis 1 (χ^2^(2) = 2.871; CFI = 0.989; NFI = 0.967; NNFI = 0.968; SRMR = 0.035); that is, the neuroticism personality trait (exogenous variable), measured at T1, T2, and T3, positively predicts psychological harassment (endogenous variable) at T3 (Figure 1). However, emotional and behavioral coping styles did not show a good fit in predicting harassment (Table 2); as can be observed, CFI, NFI, and NNFI are below 0.90, and SRMR is above 0.06.

## 4. Discussion

The results of this study confirm that the neuroticism personality trait predicts psychological harassment in a 6-year longitudinal study from the beginning of nurses’ training until three years after their professional role begins.

Based on our findings, 8.8% of Spanish nurses perceived themselves as being harassed at least once a week for six months. A meta-analysis carried out on 86 independent samples found an average prevalence of workplace bullying of 14.6% [30]. This higher prevalence could be explained by the different instruments and sampling methods used in their meta-analytic study. In fact, they identified three different assessment procedures. The procedure that is similar to ours, the behavioral experience measure without a given definition, resulted in a bullying prevalence of 11.3%, which is lower than the average prevalence obtained in the meta-analysis [30], but higher than our 8.8%. It is possible that applying Leymann’s criterion applied to our sample (being harassed at least once a week for six months) resulted in the more restrictive prevalence rate in our study. Furthermore, in a sample of Greek nurses [9], a prevalence of psychological harassment of 30.2% was reported. In this case, this percentage was the result of adding together the four alternatives for the 23rd item “Have you been bullied at work?”: b (rarely: 17.2%), c (occasionally: 9.9%), d (a few times per week: 2.0%), and e (almost daily: 1.1%). In our study, we used only a yes/no alternative, and the yes alternative (frequency of at least one harassment behavior per week for a minimum period of 6 months) produced a lower prevalence because the alternatives were not added together. However, in a review on bullying across different occupations [8], the prevalence was quite similar to ours (between 9% and 15%). 

Neuroticism was also associated with harassment. Our literature review detected only one similar study in Slovak nurses [31], but the authors did not find any relationship between neuroticism and mobbing (although they used their own mobbing questionnaire). When we looked at this relationship in other types of samples, we found contradictory results. For instance, no significant correlations were found in Australian university students [32] or Norwegian employee samples [12]. However, in a non-managerial sample of Italian employees, a significant correlation was obtained between neuroticism and bullying [6]. This disparity can be attributed to different conceptualizations of the phenomenon, different instruments used to measure personality and harassment, and the wide variety of samples. We concluded that neuroticism needs further examination through longitudinal harassment studies in the nursing profession.

As we established in our hypotheses, the neuroticism trait was relevant in explaining nursing harassment six years later. This is a very interesting result with no supporting evidence from longitudinal nursing studies in the literature. The potential explanation for this predisposition to harassment in highly neurotic nurses could be their extreme hyperreactivity to stressors. Based on scientific evidence, highly emotional nurses are more sensitive to signs of punishment, perceive events as more dramatic and uncontrollable, and react in an exaggerated emotional way, making their own problematic situations worse [33]. These extreme emotional reactions from this group of nurses could provoke more harassment behaviors from their co-workers [12].

This study has some limitations. The first is the reduction in the sample size from wave one to wave three. Professional nurses with high levels of harassment would have been more motivated to participate in the last phase of our study. However, the low participation in wave three could be due to feelings of embarrassment and shame produced by the possibility of being labelled a victim of harassment or a highly emotional reactive professional. Nevertheless, the three-wave assessment from the beginning of nursing training to the professional nursing role can be considered an important strength. Follow-up studies are needed with larger professional nursing samples, following them for several years at work, in order to replicate our study and consider the possible reverse causality hypothesis (e.g., psychological harassment predicts high neuroticism levels in nursing staff).

## 5. Conclusions

Psychological harassment is a serious work-stress problem in nursing. Based on our results, this phenomenon can be predicted by emotional instability (neuroticism). These findings have interesting implications for professional practice and nurses’ wellbeing. Based on our findings, and in order to prevent negative effects of harassment and work stress in nursing, proactive interventions are needed, not only in the form of social support from colleagues and staff, but also through training in emotional self-control techniques, such as relaxation or mindfulness, in both nursing degrees and postgraduate studies.

## Figures and Tables

**Figure 1 ijerph-16-00889-f001:**
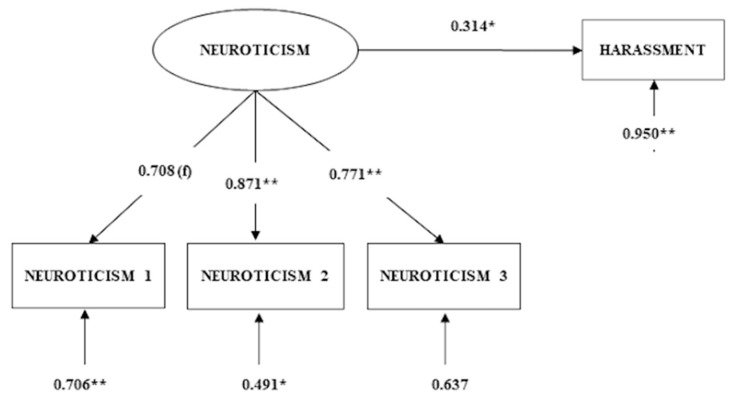
Results of structural equation modelling between neuroticism and harassment. Standardized solution. Note: * *p* < 0.05; ** *p* < 0.01.

**Table 1 ijerph-16-00889-t001:** Descriptive statistics and correlations among the study variables (*n* T1 = 249; *n* T2 = 199; *n* T3 = 70).

Variable Number	Variable	M	SD	1	2	3	4	5	6	7	8	9
1	Neuroticism T1	21.71	7.23	-								
2	Neuroticism T2	22.07	9.10	0.63 **	-							
3	Neuroticism T3	21.53	8.58	0.52 **	0.67 **	-						
4	Emotional Coping T1	1.82	0.54	0.35 **	0.44 **	0.33 **	-					
5	Emotional Coping T2	1.90	0.55	0.27 *	0.47 **	0.34 **	0.66 **	-				
6	Emotional Coping T3	1.85	0.48	0.24 *	0.20	0.33 **	0.54 **	0.35 **	-			
7	Behavioral Coping T1	1.29	0.36	−0.07	−0.03	0.03	0.31 **	0.26 *	−0.05	-		
8	Behavioral Coping T2	1.38	0.34	0.05	0.10	−0.04	0.35 **	0.49 **	−0.11	0.59 **	-	
9	Behavioral Coping T3	1.34	0.29	−0.21	−0.21	−0.21	0.21	0.20	0.09	0.48 **	0.32 **	-
10	Harassment T3 (total score)	10.12	10.68	0.22	0.22	0.34 **	−0.02	0.26 *	−0.18	0.19	0.29 *	−0.13

* *p* < 0.05; ** *p* < 0.01.

**Table 2 ijerph-16-00889-t002:** Goodness-of-Fit Indices for All Path Models.

Model	*χ*2	*df*	CFI	NFI	NNFI	SRMR
Neuroticism-Harassment	2.871 *^ns^*	2	0.989	0.967	0.968	0.035
Emotional Coping-Harassment	12.581 *	2	0.846	0.832	0.539	0.099
Behavioral Coping-Harassment	12.581 *	2	0.899	0.876	0.698	0.080

Note: *df* = Degrees of Freedom; CFI = Comparative Fit Index; NFI = Normed Fit Index; NNFI = Non-Normed Fit Index; SRMR = Standardized Root Mean Square Residual; *ns* = Non-significant; * *p*-value < 0.05.

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
