# Peer review of "The Role of Neuroticism in Predicting Psychological Harassment in Nursing: A Longitudinal Study"

_ijerph, 2019, doi:10.3390/ijerph16050889_

Round 1

Reviewer 1 Report

The manuscript describes a longitudinal study where is investigate whether nursing students’ neuroticism trait and coping styles can predict psychological harassment in nurses at work.

The study addresses the gap of the lack of longitudinal studies analyzing the influence of personality on psychological harassment.

The study is clearly described, the method and the results are appropriately presented, and the conclusions are based in the results.

Some minor changes to be considered are:

Some aspect of the introduction can be extended. For example, the importance of personality in relation with different behaviors can be mentioned (e.g., Ajzen, 2005; Hathaway & Monachesi, 1963; Saiz, Alvaro & Martínez, 2011).

Information of the initial sample should be included.

Examples of items for all the scales used in the should be given

References

Ajzen, I. (2005). Attitudes, personality, and behavior. McGraw-Hill Education (UK).

Hathaway, S. R., & Monachesi, E. D. (1963). Adolescent personality and behavior.

Saiz, J., Álvaro, J. L., & Martínez, I. (2011). Relación entre rasgos de personalidad y valores personales en pacientes dependientes de la cocaína. Adicciones, 23(2), 125-132.

Author Response

Thank you for your review and comments. Information related to personality and unhealthy behaviors, data from the initial sample of the study, and examples of items from the different scales of measurement have been provided.  

Reviewer 2 Report

Dear authors, I have read your manuscript with great interest. You have made a big job - congratulations.

 The conduct and design of the study is of reasonable quality and the presentation of the results is also reasonable. The discussion of the results is clear.

However there are some limitations of your paper:

Material and methods

The authors should specify during what years the study has been carried out, it is true that in the abstract they indicate that it is prospective for 6 years, but then there are three different courts, the 3 cuts belong to the same year? Is it a court every year? and to what years the courts correspond, should they clarify it.

The study has passed an ethics committee? if so, they should be indicated.

In the discussion made in lack of comparison with more current results, although one of the appointments  is from 2016 the other two are from 2005 and 2010, I do not know if the authors have not found more current studies, even so the results of this study will come to alleviate this situation. Very good job.

The article needs minor changes in the English revision.

These improvements can make the excellent work highly recommended for publication.

Author Response

Thank you for your review and comments. Information related to ethic committee and the different cohorts have been provided.  

Reviewer 3 Report

Article is well written and easy to understand. The only recommendations are listed below.

-Write percent instead of using symbol (%) in the following lines:

21, 44, 46, 132, 134, 139, 142-144. 148.  The symbol is used in graphs, tables and formulae.

-Line 44 enter work related stress instead of work-stress.

-Line 127 provide brief statement about Table 2 illustrations following the table.

-Lines 168 and 167: single digit numbers should be written, one and three. Use Arabic for double digit numbers. 

Author Response

Thank you for your review and comments. All changes suggested have been made.